# Managing Vascular Pedicle Exposure in Free Tissue Transfer Using a Reprocessed Micronized Dermal Substitute in Lower Extremity Reconstructions

**DOI:** 10.3390/bioengineering11030241

**Published:** 2024-02-28

**Authors:** Daheui Kim, Jun Hyeok Lee, Min Suk Park, Ma Rhip Ahn, Daiwon Jun, Jung Ho Lee

**Affiliations:** Department of Plastic and Reconstructive Surgery, Bucheon St. Mary’s Hospital, College of Medicine, The Catholic University of Korea, Seoul 14647, Republic of Korea; daheui.kim.92@gmail.com (D.K.); saylee1231@naver.com (J.H.L.); camdj@naver.com (M.S.P.); newmr0206@naver.com (M.R.A.); tfm0822@catholic.ac.kr (J.H.L.)

**Keywords:** free tissue transfer, pedicle exposure, reprocessed micronized dermal substitute, lower extremity reconstruction

## Abstract

Lower extremity reconstruction is challenging because of its intricate anatomy and dynamic biomechanics. Although microsurgical free tissue transfer offers pivotal solutions to limited local tissue availability, vascular pedicle exposure after free tissue transfer is common. We evaluated a novel method of managing pedicle exposure after free tissue transfer using a reprocessed micronized dermal substitute. Ten patients who underwent lower-extremity reconstruction using free tissue transfer and micronized dermal substitute between January and December 2023 were retrospectively reviewed. When native tissue could not be closed over the pedicle, reprocessed micronized artificial dermal matrix (rmADM) was cut and stacked to protect and stabilize it. Epithelialization was achieved by secondary skin grafting or healing by secondary intention. Flap dimensions, recipient artery and vein, ADM size, time required for granulation tissue maturation and complete epithelialization, and flap outcomes were analyzed. The mean age was 55.80 ± 20.70 years, and six patients (60%) were diabetic. The mean rmADM coverage area was 8.70 ± 8.41 cm^2^, and the average time required for complete epithelialization was 50.89 ± 14.21 days. Except for one total necrosis due to bypass graft failure, nine limbs were successfully salvaged. Application of rmADM offers numerous advantages, including vascular collapse prevention, moisture maintenance, granulation tissue growth promotion, and pedicle stabilization.

## 1. Introduction

Reconstructive plastic surgery, particularly lower extremity reconstruction, presents a multifaceted challenge for surgeons. The intricate nature of the lower limb anatomy, coupled with dynamic biomechanics, demands an understanding of surgical techniques and patient-specific considerations. The limited availability of native tissue in the lower extremities poses an additional obstacle, requiring a delicate balance between aesthetic and functional outcomes [1]. Addressing these complexities requires the fusion of technical expertise and adaptability in diverse clinical scenarios.

Microsurgical free tissue transfer provides a pivotal solution to the challenges posed by the limited availability of local tissue [2,3]. This technique not only provides a robust source of well-vascularized tissue, but enables the recreation of intricate three-dimensional structures essential for optimal functional and aesthetic outcomes. The versatility of free tissue transfer is invaluable in cases where traditional methods fail, offering a transformative approach to overcoming the hurdles presented by the intricate anatomy and biomechanics of the lower limbs. By harnessing the benefits of free tissue transfer, reconstructive plastic surgeons can achieve remarkable success in restoring form and function, significantly enhancing the overall effectiveness of lower-extremity reconstruction [4].

Another recent advancement in reconstructive surgery is the introduction of artificial dermal matrix (ADM) [5,6,7]. Serving as a biological scaffold in the wound healing process, it facilitates neovascularization, fibroblast infiltration, and improved re-epithelialization [8,9]. Its efficacy has been demonstrated in diverse clinical areas, including craniofacial, breast, burn, trauma, and diabetic foot reconstruction [10]. Consequently, ADM is one of the most extensively used skin substitutes for wound care. The aim of this study is to assess the feasibility of a novel method using a reprocessed micronized dermal substitute to manage free flap pedicle exposure in lower extremity reconstruction. The experience is analyzed and reviewed in order to assess the advantages and disadvantages of this approach.

## 2. Materials and Methods

All patients who underwent free tissue transfer to the lower extremity between January 2023 and December 2023 were screened for inclusion. Cases in which rmADM was not utilized, as well as instances where rmADM was applied in areas other than the exposed pedicle, such as the flap donor site or accompanying defects, were subsequently excluded from the analysis. The study was approved by the Institutional Review Board (HC24RISI0009) and was conducted in accordance with the Declaration of Helsinki. 

### 2.1. Materials

CGDerm matrix^®^ (CGbio Co., Seoul, Republic of Korea) is a reprocessed micronized ADM (rmADM) developed to provide optimal outcomes in skin grafting [11]. After freeze-drying, the human-derived ADM is pulverized and reprocessed into sheets with a thickness of 0.7 mm (Figure 1). Because of its nature, rmADM has several distinctive features. First, its uniform thickness provides shorter diffusion distances, resulting in lower metabolic demand and faster revascularization. Second, unlike conventional ADMs in the solid sheet form, the mesh form of rmADM facilitates superior plasmatic imbibition. Third, the concentrated collagen and extracellular matrix components are believed to provide a robust dermal structure that inhibits graft contracture. Fourth, rmADM has no polarity, and can be readily applied without distinguishing between its dermal and basement membrane sides (Figure 2).

### 2.2. Surgical Techniques

All surgeries were performed by a single senior surgeon. Depending on the location and size of the defects, various free tissue transfers were performed; these included anterolateral thigh (ALT), medial sural artery perforator (MSAP), superior gluteal artery perforator (SGAP), and superficial inferior epigastric artery perforator (SIEA) flaps. Signs of vascular compromise, such as flap color change and/or handheld Doppler sounds, were assessed after primary closure. In patients in whom vascular compression was detected, total stitch-out was performed over the flap pedicle and rmADM was applied. The rmADM was cut and stacked to stabilize and safely envelope the exposed pedicles (Figure 3). A multiporous silicone contact layer (Mepitel One, Molnlycke Health Care, Gothenburg, Sweden) and a secondary polyurethane foam dressing (Easyfoam, CGbio Co., Seoul, Republic of Korea) were used to prevent desiccation and promote granulation tissue growth. Final epithelialization was achieved using healing by secondary intention or skin grafting. 

### 2.3. Postoperative Care

Assessment of flap perfusion (e.g., color, temperature, refill, turgor, and Doppler sound) was performed every 2 h on the first postoperative day, 3 h on the second postoperative day, 4 h on the third postoperative day, and 6 h from the fourth to the seventh postoperative day. Administration of prostaglandin E1 (Eglandin, 10 μg/2 mL, IV) was maintained for 1 week and low dose aspirin (100 mg, PO) was maintained for 3 months after surgery. The antibiotic treatment was tailored to the preoperative tissue culture harvested from the defect. A standardized dangling regimen is commenced on the seventh postoperative day following a period of strict bed rest and leg elevation. Subsequent to positioning the patient at the edge of the bed, the reconstructed lower extremity is wrapped in elastic bandages and subjected to dangling sessions three times daily. The duration of each dangling session is incrementally extended by 5 min, commencing at 5 min for three days. After confirmation of flap stability and ambulatory status, the patient was discharged. The wounds were monitored in the outpatient department. If granulation tissue maturation was observed over the applied rmADM, the patient was readmitted for secondary skin grafting. In cases where signs of epithelialization were present, dressings were continued for healing by secondary intention. 

### 2.4. Measured Parameters and Outcomes

The flap dimensions, recipient artery and vein, size of ADM required to cover the pedicle, interval between primary surgery and secondary skin grafting, time required for complete epithelialization, and flap outcome were analyzed. Complete epithelization was defined as the absence of raw surfaces determined by one senior surgeon (DWJ). Flap complications were categorized as partial flap necrosis (<50%) or total flap necrosis (≥50%). 

## 3. Results

### 3.1. Patient Demographics

Ten patients were identified as eligible; their demographics are shown in Table 1. The mean age was 55.80 ± 20.70 years (range: 6–78 years). Nine patients were male, and one was female. Six patients were diabetic and five were active smokers. The most common cause of the defect was diabetic foot (n = 6), followed by trauma (n = 3), and tumor (n = 1).

### 3.2. Reconstruction Characteristics

In three patients, primary closure was possible initially, but following flap revision, tissue edema made it impossible (Table 2). In seven patients, primary closure was impossible during the initial surgery. The foot was the most common site of reconstruction (60%). The mean size of the flaps was 122.7 ± 112.32 cm^2^ (range: 10–400 cm^2^). The dorsalis pedis artery (DPA) was the most common recipient artery (40%), and the greater saphenous vein (GSV) was the most common recipient vein (30%). 

### 3.3. Treatment Outcomes

The mean area of rmADM coverage was 8.70 ± 8.41 cm^2^ (range: 2–30 cm^2^) (Table 3). The most common method of secondary coverage was split-thickness skin grafting (STSG) (60%), followed by healing by secondary intention (20%) and full-thickness skin grafting (FTSG) (10%). The average time required for maturation of granulation tissue was 30.71 ± 15.18 days. The average time required for complete epithelialization was 50.89 ± 14.21 days. Two flaps had partial necrosis. Total flap necrosis occurred in one patient in whom obstruction in the femoral-posterior tibial artery bypass graft caused arterial insufficiency, despite the intact flap anastomosis. 

### 3.4. Case Reports

#### 3.4.1. Case 1

A 70-year-old man with a history of diabetes presented at our institution with gangrenous changes in his left foot. Repeated wound failure occurred after multiple amputations, and the patient was referred for a limb-salvage operation. After surgical debridement, reconstruction was performed using a free ALT flap based on the DPA and the concomitant vein. The exposed pedicle was covered with rmADM because the primary closure of the dorsal skin resulted in pedicle compression. Three weeks after the primary surgery, the defect was filled with healthy granulation tissue and subsequent STSG was performed. One month after the STSG, the patient was able to wash and ambulate freely (Figure 4).

#### 3.4.2. Case 2

A 47-year-old man with a history of DM visited the emergency department because of an ascending infection from a diabetic foot ulcer on his right foot. Gangrene occurred on the patient’s forefoot despite abscess drainage and antibiotic treatment. A limb-salvage operation using a free SIEA flap based on the DPA and concomitant vein was performed after trans-metatarsal amputation. Venous thrombosis occurred on the first postoperative day and required GSV transposition. Because of tissue edema, the skin over the vascular pedicle did not close after primary surgery. The exposed pedicle and GSV were covered with rmADM. Six weeks after flap surgery, the defect was filled with healthy granulation tissue. After a subsequent STSG, the patient was able to ambulate (Figure 5). 

## 4. Discussion

Lower extremity reconstruction is considered one of the most challenging tasks in the reconstruction field. Numerous adverse conditions such as a lack of sufficient native tissue, high orthostatic venous pressure, concomitant peripheral arterial disease (PAD), and the potential to bear weight make optimal wound healing in extremity reconstruction a challenge. Although effective, reconstruction using free tissue transfer demands substantial time and effort. Therefore, precise pre-operative planning is necessary. One should consider the position of the flap, the direction in which the pedicle will be positioned, how the pedicle will be pulled, the amount of skin tension, motion caused by ambulation, motion of the tendon underneath, and, ultimately, the potential for the flap to bear the patient’s weight. 

Despite precise presurgical planning, the inherent laxity of native tissue in foot and distal lower extremity is often inadequate to facilitate primary closure of the pedicle [12]. This is mostly due to tissue edema that is exacerbated in lower extremity surgery secondary to underlying renal status, inflammation of the injury itself, prolonged surgery, and lymphatic disruption [13,14,15]. Such conditions are more often encountered in revision surgeries caused by vascular complications requiring microsurgical re-exploration. In our study, nine cases (90%) were located within the foot and ankle region, where the skin was thin and the surrounding soft tissue was sparse. Three cases (30%) were induced by revision surgery owing to vascular problems. 

In situations where primary closure is not possible, the conventional methods of managing exposed vascular pedicles are as follows: (1) mobilization of the flap proximally toward the pedicle, (2) promotion of healing by secondary intention, and (3) performance of a skin graft over the vascular pedicle. The mobilizing flap prioritizes the protection of the vascular pedicle and anastomosis site. However, this leads to incomplete defect coverage. In the case of implant and/or bone exposure, this option is unsuitable. The second option is to induce epithelialization by secondary intention through dressing. Although simple, it is time-consuming, and the risk of pedicle desiccation cannot be understated. Application of an STSG or FTSG to the exposed pedicle is the preferred approach. Han et al. reported a graft take rate of 99.3% using an STSG over the exposed pedicle in 15 patients without flap compromise [16]. Kovar et al. discussed the safety of skin grafts (i.e., split-andfull-thickness) over an exposed vascular pedicle [17]. No significant differences in flap outcomes (venous thrombosis, arterial insufficiency, flap loss, or dehiscence) were observed between the group in which primary closure was performed, and the group in which skin grafting was performed. The two studies were identical in that the grafts were sutured and covered with a secondary dressing only. Fixing the graft with a bolster dressing or negative-pressure wound therapy will inevitably cause pressure and increase the possibility of vascular compromise. Although both studies reported that >99% of the grafts had taken, many authors have raised questions about the effectiveness of skin graft inosculation over an exposed pedicle [18,19]. In our experience, skin grafts placed over exposed vessels function as temporary biological dressings because the graft cannot be fixed. Consequently, this necessitates a refined method for managing exposed vascular pedicles. An anecdotal fibroblast growth factor-impregnated collagen sponge has been reported to induce epithelialization without additional skin grafting [20]. Apart from preserving anastomotic patency, maintenance of pedicle position is another major concern. Spraying fibrin glue has been suggested for stabilizing the pedicle and preventing torsion [21,22]. 

Since the introduction of processed ADM implanted in a full-thickness pig wound in 1995 [23], ADMs have been used mainly for addressing full-thickness burn wounds by serving as a permanent dermal substitute. In recent years, the versatility of ADM in reconstructive surgery has revolutionized the field, providing a dynamic solution to numerous reconstruction challenges. These include managing defects involving exposed bone, tendon, and cartilage, which were traditionally deemed “ungraftable” [24]. This innovative biomaterial serves as a scaffold that supports tissue regeneration and cellular integration and provides a framework for host cell infiltration and new collagen formation. Artificial dermal matrices have various compositions (e.g., allograft versus xeno, natural versus synthetic, and single-layer versus multilayer), allowing for customization based on the specific needs of each patient and the nature of the defect being addressed [25,26,27]. As reconstructive surgeons increasingly integrate artificial dermal matrices into their armamentarium, the versatility of this technology continues to redefine the possibility of achieving optimal outcomes across a spectrum of reconstructive challenges. The use of ADM for pedicle exposure was first reported by Leclère et al. [28]. In their series of ten free-flap reconstructions of the lower extremities, Integra was used to cover the pedicle exposure, yielding no intraoperative flap complications. A comparison of outcomes between the study conducted by Leclère et al. and our own study reveals differences. Specifically, in our study, the mean area of ADM coverage was 8.70 ± 8.41 cm^2^, as opposed to 12.8 ± 2.3 cm^2^ reported by Leclère et al. Additionally, the interval between primary surgery and secondary skin grafting in our study was 4.39 ± 2.27 weeks, while Leclère et al. reported a shorter interval of 3.4 ± 0.8 weeks. We attribute these differences to a significant proportion (60%) of our cases were diabetic foot reconstructions. 

Originally developed for skin grafting [11], we found rmADM suitable for vascular pedicle coverage for the following reasons. First, the operator was free from the fear of pedicle compression. Because the product was reprocessed into a thin mesh, it was lightweight, and the risk of vascular collapse was minimal. Second, effective exudate control and prevention of pedicle desiccation could be achieved. Its mesh-like composition allows discharge from the surrounding tissue and minor leakage from the anastomotic site to be absorbed. Consequently, the retained fluid reduces the risk of pedicle desiccation and relieves the pedicle from the compression caused by a hematoma or seroma. Finally, the dermal substitute can act as a pedicle stabilizer, analogous to a fibrin spray. Because the ADM can be cut to the desired shape and stacked in multiple layers without distinguishing polarity (i.e., “dermal side” or “basement membrane side”), surgeons can handily fabricate rmADM to prevent torsion and/or kinking of the pedicle. 

A major disadvantage of this method is the time required for granulation tissue maturation and epithelialization. The average interval for granulation tissue maturation for secondary grafting was 30.71 ± 15.88 days, and the average time required for complete epithelialization was 50.89 ± 14.21 days. The outcomes of STSG in diabetic foot ulcers exhibit considerable variability due to the inherent heterogeneity among patients. A retrospective analysis of 107 diabetic foot ulcer patients treated with STSG revealed a mean time to healing of 5.1 weeks, with a range spanning from 3 to 16 weeks [29]. In contrast, a prospective case–control study involving 100 diabetic patients demonstrated a significantly shorter mean healing time of 28 ± 5 days in the graft group, compared to 122 ± 7 days in the dressing group [30]. Furthermore, a randomized prospective clinical trial utilizing living human skin equivalents reported a mean time to complete closure of 65 days [31]. While our findings align with previous research, it is noteworthy that our observed healing times may be considered delayed compared to the rapid epithelialization typically seen with STSGs in burn injuries, which often achieve closure within two weeks [32]. The impaired healing observed in diabetic patients can be attributed to a multitude of factors, including impaired macro and microcirculation, peripheral neuropathy, endothelial dysfunction, suboptimal glycemic control, and impaired neuropeptide signaling [33,34]. Another factor that may have influenced the epithelialization rate is the duration of follow-up intervals. The average follow-up interval was 8.3 days, with a range of one to 28 days, depending on patients’ economic and ambulatory statuses. We acknowledge that longer follow-up intervals may have contributed to delayed detection of complete epithelialization, thereby prolonging the epithelialization time. A second disadvantage is increased cost for adding rmADM to the surgery. The cost of rmADM (CGDerm matrix^®^) per square centimeter in Korea is approximately $30. Considering that the average size of ADM used was 8.70 ± 8.41 cm^2^, the additional cost incurred per patient amounts to approximately $261. Regarding the instance of total necrosis, thrombus formation was identified within the bypass graft distant from the anastomosis. Consequently, the likelihood of a causal relationship with the use of rmADM is deemed minimal. As for the two cases of partial necrosis, the relationship with rmADM is unclear. Authors suspect multiple factors such as tight closure, large flap size, or injury to perforators during harvest may have contributed to these occurrences. Nonetheless, nine out of ten limbs were successfully salvaged.

The limitations of this study include its retrospective nature, observational design, and small sample size. Further investigations with prospective study designs, large sample sizes, and regular follow-up intervals are warranted. However, to the best of our knowledge, this study is the first to apply rmADM to exposed vascular pedicles. We believe that this novel method could provide a valid option for the management of free-flap pedicle exposure in lower-extremity reconstruction. 

## Figures and Tables

**Figure 1 bioengineering-11-00241-f001:**
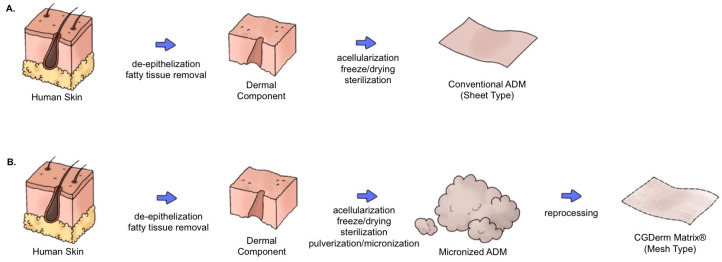
Illustration of reprocessed micronized artificial dermal matrix (rmADM) production. In contrast to conventional ADM (**A**), additional pulverization and micronization is required. The micronized ADM is reprocessed into a mesh form with a uniform thickness of 0.7 mm (**B**).

**Figure 2 bioengineering-11-00241-f002:**
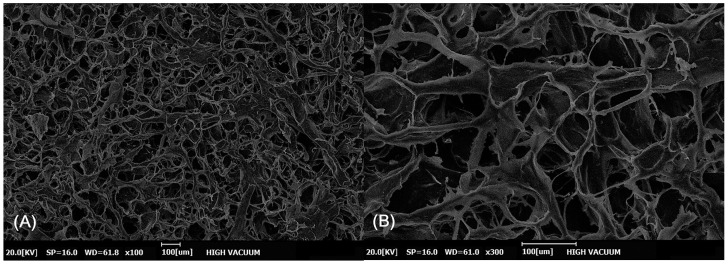
The microscopic structure of reprocessed micronized artificial dermal matrix (rmADM) viewed through a scanning electron microscope (SEM). Regardless of the micronization, the integrity of the protein scaffold in the dermis is maintained. This promotes plasma imbibition, fibroblast infiltration, and neovascularization. SEM with ×100 magnification (**A**), with ×300 magnification (**B**).

**Figure 3 bioengineering-11-00241-f003:**
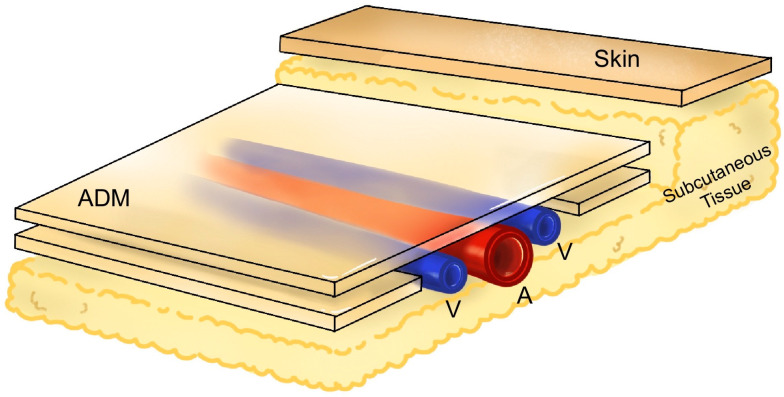
Depiction of how reprocessed micronized artificial dermal matrix (rmADM) is applied to an exposed vascular pedicle. The rmADM is cut and stacked over the vascular pedicle to fulfill the following purposes; stabilize the pedicle from kinking, protect it from desiccation, and promote granulation tissue growth.

**Figure 4 bioengineering-11-00241-f004:**
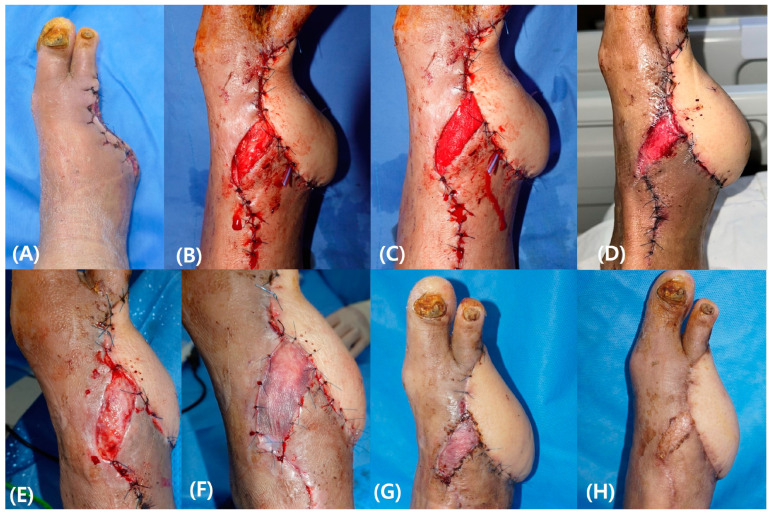
A case of 70-year-old man. Repeated wound failure occurred after multiple amputations (**A**). After reconstruction using a free anterolateral thigh flap, primary closure of the dorsal skin caused pedicle compression (**B**). The exposed vascular pedicle was covered with reprocessed micronized artificial dermal matrix (rmADM) (**C**). Growth of granulation tissue was noted one week after surgery (**D**) and maturation was noted three weeks after surgery (**E**) followed by subsequent split thickness skin graft (STSG) (**F**). The graft revealed complete take one month after STSG (**G**) and stable epithelialization enabling full rehabilitation three months after STSG (**H**).

**Figure 5 bioengineering-11-00241-f005:**
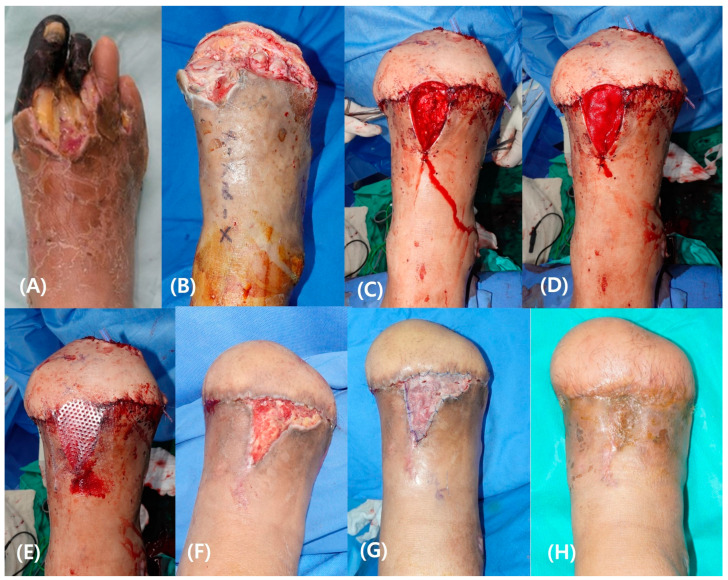
A case of 47-year-old man. Gangrene occurred on the patient’s forefoot after infection (**A**). After trans-metatarsal amputation (**B**), limb-salvaging surgery using a free superficial inferior epigastric artery flap was performed. Venous thrombosis occurred on the first postoperative day, requiring microsurgical re-exploration. Due to tissue edema, the skin over the vascular pedicle was not closed primarily (**C**). The exposed flap pedicle and GSV were covered with reprocessed micronized artificial dermal matrix (rmADM) (**D**), and multiparous silicone contact layer dressing was applied (**E**). Six weeks after flap surgery, the defect was filled with healthy granulation tissue (**F**), and a subsequent split thickness skin graft was performed (**G**). Stable epithelialization was achieved three months after the skin graft (**H**).

**Table 1 bioengineering-11-00241-t001:** Patient demographics.

Variables	Total Patients (n = 10)
Age (years)	55.80 ± 20.70 (6–78)
Sex	
Male	9
Female	1
DM	6
HTN	2
Current smoker	5
Cause of defect	
Diabetic foot	6
Trauma	3
Tumor	1

DM: diabetes mellitus, HTN: hypertension.

**Table 2 bioengineering-11-00241-t002:** Defect and flap characteristics.

N	Type of Surgery	Cause of Revision	Defect Location	Type of Flap	Flap Dimensions (cm^2^)	Recipient Artery and Vein
1	Primary		Great toe	MSAP	9 × 5	DPA & DPV
2	Primary		Foot	ALT	15 × 7	DPA & DPV
3	Revision	Venous thrombosis	Foot	SIEA	20 × 8	DPA & GSV
4	Revision	Arterial insufficiency	Great toe	RASP	2.5 × 4	FDMA & FDMV
5	Revision	Arterial insufficiency	Knee	SGAP	11 × 6	MSA & MSV
6	Primary		Ankle	SIEA	20 × 12	PTA & PTV
7	Primary		Foot	MSAP	15 × 5	ATA & ATV
8	Primary		Foot	ALT	25 × 16	PTA & GSV
9	Primary		Foot	SIEA	14 × 7	PTA & PTV
10	Primary		Foot	MSAP	7 × 4	DPA & GSV

MSAP: medial sural artery perforator flap, ALT: anterolateral thigh flap, SIEA: superficial inferior epigastric artery flap, RASP: radial artery superficial palmar branch flap, SGAP: superior gluteal artery perforator flap, DPA: dorsalis pedis artery, DPV: dorsalis pedis vein, GSV: great saphenous vein, FDMA: first dorsal metatarsal artery, FDMV: first dorsal metatarsal vein, MSA: medial sural artery, MSV: medial sural vein, PTA: posterior tibial artery, PTV: posterior tibial vein, ATA: anterior tibial artery, ATV: anterior tibial vein.

**Table 3 bioengineering-11-00241-t003:** Closure of ADM and flap outcome.

N	ADM Size (cm^2^)	Method of Coverage	Interval between ADM and Skin Grafting (Days)	Time to Complete Epithelialization (Days)	Flap Complication
1	2 × 4	STSG	9	46	None
2	2 × 5	STSG	21	44	None
3	5 × 6	STSG	42	71	None
4	1 × 2	Healing by secondary intention	NA	36	None
5	2 × 2	STSG	46	46	Partial flap necrosis
6	3 × 5	NA	NA	NA	Total flap necrosis due to bypass failure
7	2 × 3	STSG	49	70	None
8	1 × 4	Healing by secondary intention	NA	56	Partial flap necrosis
9	1 × 4	STSG	33	59	None
10	1 × 4	FTSG	15	30	None
	8.70 ± 8.41		30.71 ± 15.18	50.89 ± 14.21	

ADM: artificial dermal matrix, STSG: split thickness skin graft, FTSG: full thickness skin graft.

## Data Availability

The original contributions presented in the study are included in the article, further inquiries can be directed to the corresponding author.

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
