# Peer review of "Managing Vascular Pedicle Exposure in Free Tissue Transfer Using a Reprocessed Micronized Dermal Substitute in Lower Extremity Reconstructions"

_bioengineering, 2024, doi:10.3390/bioengineering11030241_

Round 1
Reviewer 1 Report
Comments and Suggestions for Authors
The paperwork is appropriate for the journal, covering an interesting bioengineering technique for free flap pedicle exposure, using a reprocessed micronized dermal substitute. The technique seems to be original and is interesting regarding the potential. Do the authors use a different reprocessed micronized dermal substitute – see reference 8?
The study is retrospective, has informed consent, and is board-approved, so no ethical issues were detected. The figures & tables add value to the paper, and I would cite the paper if published. Although the data provided are scarce and backed up partially by the author’s findings, the proposal will merit further investigation and eventually translation into daily practice. I don’t see fatal flaws in the paper.
1. Abstract:
Is the abstract adequate?
_ Yes
X_ No
If no, what is missing – vascular problems and topographic particularities in lower extremity reconstruction (not every defect warrants a microsurgical flap – mainly the foot and lower third of the leg).
When they say “novel” does they refer to this article - 8. Hahn, H.M.; Jeong, Y.S.; Lee, I.J.; Kim, M.J.; Lim, H. Efficacy of split-thickness skin graft combined with novel sheet-type reprocessed micronized acellular dermal matrix. BMC Surg. 2022, 22, 358. DOI:10.1186/s12893-022-01801-x.?
How do the partial necrosis cases and total necrosis case relate to this method?
2. Figures:
Are all the tables/figures/illustrations essential?
_X_Yes
No
3. References:
X Adequate
_X_ Inadequate
If inadequate, what is missing – cannot find the reference 8 in article text.
4. Conclusions:
__Warranted
_X_Partly justified - More data needed, as suggested.
Not justified—major flaw
5. Financial Disclosure/Conflict of Interest Statement given by authors
Complete; no more information needed
__ More Information/Disclosure needed – The authors state - Acknowledgments: The publication of this article was supported by CGBio, Inc. – how exactly was this done, as the ADM is produced by them. Does this influence in someway the article?
6. Is there a question about violation of the Helsinki Doctrine on Human Experimentation
X No
__Yes
7. Do you have reason to believe that this manuscript, or any significant portion thereof, has been plagiarized?
X No
__Yes
8. If published, would it warrant a discussion? If so, please name two candidates. Include yourself if appropriate.
__ No
X Yes
i._ Hahn, H.M – see reference no. 8
9. Rank this manuscript according to other manuscripts you've reviewed in the past 5 years.
__Top 10% (Must publish)
__Top 10%-20% (Good; Okay if materials needed)
__In Top 20%-30% (Good—Suggest other publication in the group)
X Below Top 30% (Not suited)
10. (IMPORTANT) INSERT CONFIDENTIAL COMMENTS TO EDITOR:
They state that is a novel technique didn’t find a similar report.
The discussion section could be improved.
Comments to Author:
Much of the discussion commentary is related to free flaps instead of this method. It also needs a grammatical some clarifications.
Comments by section:
Abstract
Content: Two flaps had partial necrosis and one had total necrosis due to bypass graft failure.
Comment; is this related to rmADM use?
Introduction
Content: Another recent advancement in reconstructive surgery is the introduction of artificial dermal matrix (ADM).
Comment: reference?
Content: In this article, we propose a novel method using a reprocessed micronized dermal substitute to manage free-flap ped- icle exposure in lower extremity reconstruction. We present our experiences and review the advantages and disadvantages of this novel approach.
Comment: The aim of this study is to assess the feasibility …. The experience is analyzed and reviewed in order to…
Methods
Content: Signs of vascular compromise such as flap color change and/or handheld Doppler sounds were assessed after primary closure.
Comment: is this done according to a free flap monitoring protocol or related to the rmADM use solely?
Content: The dangling procedure
Comment: protocol?
Content: Patients who underwent lower-extremity reconstruction using free tissue transfer and micronized dermal substitute to cover the exposed pedicle between January and De- cember 2023 were retrospectively reviewed.
Comment: what factors/variables were looked for? Inclusion/exclusion criteria?
Results
Content: Ten patients were enrolled in the study.
Comment: were identified as eligible, because they were not “enrolled” – the study is retrospective.
Discussion
Content: Although powerful, reconstruction using free tissue transfer requires consid- erable time and effort.
Comment: rephrase / powerful?
Content: pedicle exposure is common after free tissue transfer
Comment: define common?
Content: Although simple, it is time-consuming, and the risk of pedicle desiccation cannot be undermined.
Comment: rephrase undermined? Maybe Understated?
Content: In our experience, skin grafts placed over ex- posed vessels function as temporary biological dressings because the graft cannot be fixed.
Comment: the flaps that have dehiscence cannot be used for a few stitches?
Comment: The versatility of ADM in reconstructive surgery has revolutionized the field, offer- ing a dynamic solution to a myriad of reconstruction challenges.
Comment: can you elaborate? How does the ADM has revolutionized the reconstructive surgery? ADM has put on table some solutions but with a lot of problems complications? Put a side a lot of pff label uses, that can bring potential legal issues?
Content: The mean area of ADM coverage was 12.8±2.3 cm2 and the interval between primary surgery and secondary skin grafting was 3.4±0.8 weeks.
Comment: can you compare with your results? Seems a little bit different.
Content: The average interval for granulation tissue maturation for secondary grafting was 30.71±15.88 days and the average time required for complete epithelialization was 50.89±14.21 days. Delayed healing is believed to have been caused by comorbidities such as diabetes and PAD.
Comment: can you detail further? Compare with direct skin grafting?
Content: Except for one case in which obstruction in the femoral-posterior tibial artery bypass graft caused total necrosis of the flap despite anastomotic patency, no vascular complications were noted after rmADM application.
Comment: Do you believe that here is a direct causality? This obstruction will lead to a limb fail also.
Comments on the Quality of English LanguageSome rephrasing and terms need proofreading.
Author Response
Reviewer 1
Review report form
Abstract
Comment 1; vascular problems and topographic particularities in lower extremity reconstruction (not every defect warrants a microsurgical flap – mainly the foot and lower third of the leg).
When they say “novel” does they refer to this article - 8. Hahn, H.M.; Jeong, Y.S.; Lee, I.J.; Kim, M.J.; Lim, H. Efficacy of split-thickness skin graft combined with novel sheet-type reprocessed micronized acellular dermal matrix. BMC Surg. 2022, 22, 358. DOI:10.1186/s12893-022-01801-x.?
How do the partial necrosis cases and total necrosis case relate to this method?
Response 1: The abstract has been modified for clarity and to highlight the results.
Indeed, the rmADM utilized in both studies is identical. However, the novelty of our study resides in the innovative surgical technique employed, rather than the product itself. We have included a discussion concerning the potential correlation between the use of rmADM and flap necrosis within our manuscript.
References
Comment 2: cannot find the reference 8 in article text
Response 2: Reference 8 is cited within the methodology section of the manuscript. We have also cited reference 8 in discussion.
Financial Disclosure/Conflict of Interest Statement given by authors
Comment 3: The authors state - Acknowledgments: The publication of this article was supported by CGBio, Inc. – how exactly was this done, as the ADM is produced by them. Does this influence in someway the article?
Reponse 3: The rmADM (CGDerm matrix®) used in this study is manufactured by CGBio, Inc. and is commercially available in South Korea. All products employed in the study were purchased by the patients themselves. Following a retrospective review conducted by the authors, CGBio provided supplementary product information and extended support in processing the article. Apart from these disclosures, all authors declare no conflicts of interest.
Comments to Author
Abstract
Content: Two flaps had partial necrosis and one had total necrosis due to bypass graft failure.
Comment 1; is this related to rmADM use?
Response 1: Thank you for raising this important point. The authors acknowledge the necessity for further elaboration, considering the focus of our study on evaluating the feasibility and safety of a novel microsurgical technique. Regarding the instance of total necrosis, it is noteworthy that thrombus formation was identified within the bypass graft distant from the anastomosis. Consequently, the likelihood of a causal relationship with the use of rmADM is deemed minimal. As for the two cases of partial necrosis, the relationship is unclear. The authors suspect multiple factors such as tight closure, large flap size, or injury to perforators during harvest may have contributed to these occurrences. We intend to discuss this aspect in the discussion section of our paper (page 10).
Introduction
Content: Another recent advancement in reconstructive surgery is the introduction of artificial dermal matrix (ADM).
Comment 2: reference?
Response 2: According to Kamolz et al., the utilization of acellular dermal matrices (ADM) represents the reconstructive clockwork for the surgical landscape of the 21st century. Similarly, Jefferson et al., identify the application of ADM as one of the pivotal advancements in tissue engineering that has revolutionized plastic surgery. Furthermore, Nahabedian, Maurice Y., contends that the integration of ADM stands as a cornerstone advancement in prosthetic breast reconstruction. We will incorporate these works into our manuscript to substantiate our arguments effectively (page 2).
Content: In this article, we propose a novel method using a reprocessed micronized dermal substitute to manage free-flap ped- icle exposure in lower extremity reconstruction. We present our experiences and review the advantages and disadvantages of this novel approach.
Comment 3: The aim of this study is to assess the feasibility …. The experience is analyzed and reviewed in order to…
Response 3: Thank you for your suggestion. We have revised the sentence accordingly (page 2).
Methods
Content: Signs of vascular compromise such as flap color change and/or handheld Doppler sounds were assessed after primary closure.
Comment 4: is this done according to a free flap monitoring protocol or related to the rmADM use solely?
Response 4: Listed above are routine free flap monitoring protocols.
Content: The dangling procedure
Comment 5: protocol?
Response 5: A dangling procedure represents a technique aimed at progressively acclimating the flap to gravitational stresses. We have adopted and modified a method outlined by Kolbenschlag in 2015. A standardized dangling regimen is commenced on the seventh postoperative day following a period of strict bed rest and leg elevation. Subsequent to positioning the patient at the edge of the bed, the reconstructed lower extremity is wrapped in elastic bandages and subjected to dangling sessions three times daily. The duration of each dangling session is incrementally extended by 5 minutes, commencing at 5 minutes for three days. These details have been integrated into the methodology section of our manuscript (page 4).
Content: Patients who underwent lower-extremity reconstruction using free tissue transfer and micronized dermal substitute to cover the exposed pedicle between January and De- cember 2023 were retrospectively reviewed.
Comment 6: what factors/variables were looked for? Inclusion/exclusion criteria?
Response 6: All patients who underwent free tissue transfer to the lower extremity between January 2023 and December 2023 were initially screened for inclusion. Among the initial 34 cases, 10 cases in which rmADM was not utilized were excluded. Additionally, 14 cases were excluded due to the application of rmADM in areas other than the exposed pedicle, such as the flap donor site or accompanying defects. Consequently, 10 patients were eligible for the study. We have duly revised our manuscript to reflect these adjustments (page 2).
Results
Content: Ten patients were enrolled in the study.
Comment 7: were identified as eligible, because they were not “enrolled” – the study is retrospective.
Response 7: We have revised the sentence (page 4).
Discussion
Content: Although powerful, reconstruction using free tissue transfer requires considerable time and effort.
Comment 8: rephrase / powerful?
Response 8: We have rephrased the sentence into “Although effective, reconstruction using free tissue transfer demands substantial time and effort.” (page 8)
Content: pedicle exposure is common after free tissue transfer
Comment 9: define common?
Response 9: There is a paucity of literature addressing the precise frequency of pedicle exposure occurrences in free tissue transfer procedures. Unfortunately, data to quantify this frequency are not readily available. We have rephrased the sentence to “The inherent laxity of native tissue in foot and distal lower extremity is often inadequate to facilitate primary closure of the pedicle.” (page 8)
Content: Although simple, it is time-consuming, and the risk of pedicle desiccation cannot be undermined.
Comment 10: rephrase undermined? Maybe Understated?
Response 10: Thank you for your valuable suggestion. We will revise the sentence in accordance with your recommendation (page 9).
Content: In our experience, skin grafts placed over ex- posed vessels function as temporary biological dressings because the graft cannot be fixed.
Comment 11: the flaps that have dehiscence cannot be used for a few stitches?
Response 11: Primary closure remains the preferred option; however, additional stitches may potentially exert pressure on the pedicle, leading to microvascular compromise.
Comment: The versatility of ADM in reconstructive surgery has revolutionized the field, offer- ing a dynamic solution to a myriad of reconstruction challenges.
Comment 12: can you elaborate? How does the ADM has revolutionized the reconstructive surgery? ADM has put on table some solutions but with a lot of problems complications? Put a side a lot of pff label uses, that can bring potential legal issues?
Response 12: Since the introduction of processed ADM implanted in a full-thickness pig wound in 1995, ADMs have been used mainly for addressing full-thickness burn wounds by serving as a permanent dermal substitute. In recent years, the utilization of ADM has further expanded to encompass the management of complex wounds involving exposed bone, tendon, and cartilage, traditionally deemed ungraftable. This perspective aligns with Janis et al., advocating for the incorporation of reconstructoin method using dermal matrices into a revised reconstructive ladder. While ADMs are theoretically acellular and non-immunogenic, potential complications are engraftment rates and cost. While we acknowledge the ethical debates surrounding the use of human tissues, we aim to maintain focus on the objectives of this paper without deviation. We intend to refine this statement and provide pertinent references to fortify our manuscript (page 9).
Content: The mean area of ADM coverage was 12.8±2.3 cm2 and the interval between primary surgery and secondary skin grafting was 3.4±0.8 weeks.
Comment 13: can you compare with your results? Seems a little bit different.
Response 13: A comparison of outcomes between the study conducted by Leclère et al. and our own study reveals differences. Specifically, in our study, the mean area of ADM coverage was 8.70±8.41 cm2, as opposed to 12.8±2.3 cm2 reported by Leclère et al. Additionally, the interval between primary surgery and secondary skin grafting in our study was 4.39±2.27 weeks, while Leclère et al. reported a shorter interval of 3.4±0.8 weeks. We attribute these differences to a significant proportion (60%) of our cases were diabetic foot reconstructions (page 9).
Content: The average interval for granulation tissue maturation for secondary grafting was 30.71±15.88 days and the average time required for complete epithelialization was 50.89±14.21 days. Delayed healing is believed to have been caused by comorbidities such as diabetes and PAD.
Comment 14: can you detail further? Compare with direct skin grafting?
Response 14: The outcomes of split-thickness skin grafts (STSG) in diabetic foot ulcers exhibit considerable variability due to the inherent heterogeneity among patients. A retrospective analysis of 107 diabetic foot ulcer patients treated with STSG revealed a mean time to healing of 5.1 weeks, with a range spanning from three to 16 weeks. In contrast, a prospective case-control study involving 100 diabetic patients demonstrated a significantly shorter mean healing time of 28 ± 5 days in the graft group, compared to 122 ± 7 days in the dressing group. Furthermore, a randomized prospective clinical trial utilizing living human skin equivalents reported a mean time to complete closure of 65 days. While our findings align with previous research, it is noteworthy that our observed healing times may be considered delayed compared to the rapid epithelialization typically seen with STSGs in burn injuries, which often achieve closure within two weeks. The impaired healing observed in diabetic patients can be attributed to a multitude of factors, including impaired macro and microcirculation, peripheral neuropathy, endothelial dysfunction, suboptimal glycemic control, and impaired neuropeptide signaling (page 10).
Content: Except for one case in which obstruction in the femoral-posterior tibial artery bypass graft caused total necrosis of the flap despite anastomotic patency, no vascular complications were noted after rmADM application.
Comment 15: Do you believe that here is a direct causality? This obstruction will lead to a limb fail also.
Response 15: Obstruction in the bypass graft resulted both flap necrosis and distal limb necrosis necessitating subsequent below knee amputation. We believe that the outcome has minimal causality to using rmADM. We have made necessary modifications in our manuscript to clarify that matter (page 10).
Reviewer 2 Report
Comments and Suggestions for Authors
Comments:
- 1- The abstract requires more detail, particularly in highlighting significant results.
- 2- Ensure that the labels (A and B) are incorporated into Figure 2.
- 3- Enhance the dimensions and resolution of Figure 3.
- 4- The captions of the tables are currently concise; consider providing more comprehensive explanations for better clarity.
Author Response
Reviewer 2
Comment 1: The abstract requires more detail, particularly in highlighting significant results.
Response 1: The abstract has been improved by incorporating key highlights from the results (page 1).
Comment 2: Ensure that the labels (A and B) are incorporated into Figure 2.
Response 2: Thank you for point that out. Adjustments have been implemented in Figure 2 (page 3).
Comment 3: Enhance the dimensions and resolution of Figure 3.
Response 3: Adjustments have been implemented in Figure 3 with increased visibility (page 3).
Comment 4: The captions of the tables are currently concise; consider providing more comprehensive explanations for better clarity.
Response 4: Adjustments have been implemented in Table 2 and 3 (page 5 and 6).
Reviewer 3 Report
Comments and Suggestions for Authors
I enjoyed reading this article presented by Kim et al., where they describe a case series of 10 patients with exposed vessels after free flap to the foot/ankle, managed with rmADM. This is the first study investigating this dermal matrix for this purpose, and the outcomes are promising.
A few comments for improvement:
1) Methods - what was the authors definition of "time required for complete epithelialization" and how often did they see the patient in follow-up? For example, if you saw patients every 2 weeks as opposed to every week, it could be possible that time to re-epithelialization is shorter than the mean of 50 days you reported.
2) Methods - have you applied rmADM directly over anastomoses, or only over the pedicles?
3) Discussion - the authors should address the additional cost of adding rmADM to the surgery
Author Response
Reviewer 3
Comment 1: Methods - what was the authors definition of "time required for complete epithelialization" and how often did they see the patient in follow-up? For example, if you saw patients every 2 weeks as opposed to every week, it could be possible that time to re-epithelialization is shorter than the mean of 50 days you reported.
Response 1: The authors define complete epithelialization as the absence of detectable raw surfaces, a determination made by a senior surgeon (DWJ). The average follow-up interval was 8.3 days, ranging from one to 28 days depending on patients' economic and ambulatory statuses. We acknowledge that longer follow-up intervals might have influenced the time required for re-epithelialization. We intend to discuss this aspect in the method (page 4) and discussion (page 10) section of our paper.
Comment 2: Methods - have you applied rmADM directly over anastomoses, or only over the pedicles?
Response 2: Yes, in both scenarios. Regardless of the site of anastomosis, any exposed pedicles that could not be adequately covered with native tissue were addressed by utilizing rmADM.
Comment 3: Discussion - the authors should address the additional cost of adding rmADM to the surgery
Response 3: The cost of rmADM (CGDerm matrix®) per square centimeter in Korea is approximately $30. Considering that the average size of ADM used was 8.70±8.41 cm2, the additional cost incurred per patient amounts to approximately $261. We will incorporate this pertinent information into discussion section (page 10).